# SARS, MERS and COVID-19-Associated Renal Pathology

**Hristo Popov** [1],*****, **George S. Stoyanov** [1], **Lilyana Petkova** [1], **Dimo Stoyanov** [2], **Martin Ivanov** [2] **and Anton B. Tonchev** [2]

1. Department of General and Clinical Pathology, Forensic Medicine and Deontology, Faculty of Medicine, Medical University—Varna "Prof. Dr. Paraskev Stoyanov", 9002 Varna, Bulgaria
2. Department of Anatomy and Cell Biology, Faculty of Medicine, Medical University—Varna "Prof. Dr. Paraskev Stoyanov", 9002 Varna, Bulgaria
* Correspondence: popov12@abv.bg

**Definition:** Coronaviruses are a large group of RNA viruses, the most notable representatives of which are SARS-CoV, MERS-CoV and SARS-CoV-2. Human coronavirus infections were first documented in the 1960s, when members causing seasonal common colds were successfully replicated in human embryonal trachea and kidney cell cultures and classified based on electron microscopy. The history of coronaviruses stretched far back to that point, however, with some representatives causing disease in animals identified several decades prior and evolutionary data pointing towards the origin of this viral group more than 55 million years ago. In the short time period of research since they were discovered, coronaviruses have shown significant diversity, genetic peculiarities and varying tropism, resulting in the three identified causative agents of severe disease in humans—SARS, MERS and the most recent one, COVID-19, which has surpassed the previous two due to causing a pandemic resulting in significant healthcare, social and political consequences. Coronaviruses are likely to have caused pandemics long before, such as the so-called Asian or Russian influenza. Despite being epitheliotropic viruses and predominantly affecting the respiratory system, these entities affect multiple systems and organs, including the kidneys. In the kidneys, they actively replicate in glomerular podocytes and epithelial cells of the tubules, resulting in acute kidney injury, seen in a significant percentage of severe and fatal cases. Furthermore, the endothelial affinity of the viruses, resulting in endotheliitis, increases the likelihood of thrombotic microangiopathy, damaging the kidneys in a two-hit mechanism. As such, recently, COVAN has been a suggested nomenclature change indicating renal involvement in coronavirus infections and its long-lasting consequences.

**Keywords:** SARS (CoV); MERS (CoV); SARS-CoV-2; COVID-19; COVAN; nephropathology; coronavirus-associated nephritis; multisystem infection

## 1. Introduction

Coronaviruses are representatives of a large group of RNA viruses [1]. According to their structure, these viruses have relatively large sizes, varying between 80 and 120 μm, but some members are characterized by smaller and significantly larger sizes—from 50 to 400 μm, with a molecular mass of about 40,000 kilodaltons [1–4]. The viral single-stranded RNA is enveloped by a double lipid membrane in which transmembrane and structural proteins are integrated, including the so-called spike proteins [5]. Formed in this way, the virion has a characteristic rayed surface, visible in electron microscopy and computer-modeled reconstructions [1]. From this characteristic surface structure of the virions, they derive their name—*corona* viruses [1,6]. It is through these membrane-associated proteins that the virions bind to cell receptors, during which the viral endocytosis in the host cell takes place. Immediately after virion endocytosis, viral "undressing" occurs in the host cell's cytoplasm and viral RNA is released [7]. The structure of this RNA, with a 5′ methylated cap and a 3′ polyadenylated tail, allows it to be recognized by the granulated endoplasmic

reticulum as an mRNA and thus initiate direct replication of new structural proteins for the assembly of new virions, by synthesizing a replication–transcription complex [6].

The replication–transcription complex allows the viral RNA to be replicated in multiple steps, the viral structures to be transcribed and, in the presence of at least two viral RNAs, even from another virus family, in the host cell, to recombine the sources [6]. Errors in the first two described stages and the third stage of the process give the characteristics of high mutagenicity and emergence of new virus variants; a critical clarification is that such mutant and/or recombined forms are not always vital [8]. Furthermore, the hijacking of the host cell not only disrupts its metabolism and homeostasis, leading to induced apoptosis or necrosis, but can also lead to structural, functional, or mutation alterations occurring in it [9].

After the synthesis of the necessary genetic and structural material, the process of virion assembly begins, taking place predominantly in the Golgi complex, after which new virions are released from the cell by mediated exocytosis and/or cell destruction (necrosis) and can directly infect other cells [6].

Coronaviruses exhibit epithelial tropism, with individual members having specific tissue, organ and even species tropism, as well as different modes of transmission [10]. The most common infection transmission mechanism is airborne, while there are also data on fecal–oral transmission (alimentary)—coronavirus gastroenteritis in pigs [6,10–12]. Given the diversity of structural proteins, the cellular receptors used for endocytosis vary between different entities, with the most commonly used ones being the ACE-2 and alanine aminopeptidase receptors [6,13].

As a group of viruses, they cause diseases in a large proportion of mammals, including humans, as well as in birds, i.e., are a typical example of anthropozoonosis [6,14,15]. Characteristic of the course of these infections is the involvement of the respiratory system, with the majority of representatives involving the upper respiratory tract and causing seasonal common colds with acute, predominantly serous rhinosinusitis, conjunctivitis, otitis, pharyngitis and laryngitis [14,16]. Classically, these infections are not severe; they are transient, no specific treatment is available, and if necessary, the treatment is symptomatic— nasal decongestants, antipyretics and vitamins [16].

The earliest documented data on coronavirus infections date back to the 1920s, when they were described as causing severe bronchitis in newly hatched chicks, with extremely high mortality ranging from 40% to 90% [15,17]. Newly hatched chicks had severe respiratory symptoms, and the isolated virus was named infectious bronchitis virus in 1933 and cultivated in 1937 [18,19]. In the 1940s, viral infections in mice causing encephalitis and hepatitis were also described, and at a later stage, it was understood that the three viruses described so far belong to the same group [4,20]. Human coronaviruses were described in the 1960s as the causes of common seasonal colds, in which it was impossible to cultivate the causative agent using conventional methods used for adenoviruses and rhinoviruses [21,22]. Cultivation was achieved only a few years later using tissue cultures from human embryonal trachea [23]. Descriptions of similar viruses followed in the same decade, some of which were replicated successfully in human embryonal kidney cell cultures [23–25]. Utilizing electron microscopy, it was established that all the viral pathogens described so far have a characteristic shape and membrane protrusions, which is why they were united in a joint group called coronaviruses, which later included many more representatives, based on genetic analysis, with varying organ tropism and severity of symptoms [25,26].

Evolutionarily, the earliest established ancestor of modern coronaviruses is thought to have existed about 10,000 years ago, with some evidence suggesting the presence of similar viruses about 55 million years ago in bats and birds [27]. This evolutionary theory of the origin of modern coronaviruses is well supported by the fact that the natural reservoir and source of new variants are most often bats and birds [28].

The modern classification of viruses defines these viruses in the family *coronaviridae*, with two subfamilies, *letovirinae* and *orthocoronavirinae*, of which the characteristics and entities described so far are representatives [10,29]. The *orthocoronavirinae* subfamily, in

turn, consists of four *genera*—alpha, beta, gamma and delta coronavirus [27]. Alpha- and betacoronaviruses have the greatest infectious affinity for the human population, while gamma and delta are primarily zoonotic infections [10,29–31]. Betacoronaviruses are of fundamental importance to the human population, whose representations include the three most severe infections with such viruses—SARS, MERS and COVID-19 [12,32,33]. It is important to note that while the betacoronavirus family is a close relative, SARS-CoV and SARS-CoV-2 are members of the *subgenus sarbecovirus*, while MERS is a more distant relative—a member of the *subgenus* of *merbecovirus*.

Although the clinical picture in humans is relatively mild, except for some virus types, coronaviruses are thought to have caused epidemic outbreaks of severe disease long before they were identified, most often interpreted as influenza infections [6,14,16,34,35]. For example, the last major pandemic of the 19th century—the so-called Asian or Russian influenza, which caused a pandemic outbreak in 1889–1890, long considered to be influenza type 1—had similar symptoms to COVID-19, including neurological symptoms—loss of taste and smell, clouding of consciousness and encephalitis-like symptoms, all rarely seen in influenza [14,36,37]. Of course, serological data from the period do not exist, and proving a phylogenetic relationship is practically impossible.

Since emerging in late 2019, the most recent representative of severe coronaviruses, SARS-CoV-2 has led to multiple consequences, from medical and social to political and philosophical [38–41]. The clinical disease the virus causes, coronavirus disease, identified in 2019 (COVID-19), while initially regarded as a respiratory infection, has shown its multisystem nature [42,43]. While most cases present with and have the most severe clinical symptoms from the respiratory system, case reports, small cohort studies and systemic research have genuinely shown the systemic nature of the virus due to its epitheliotropism and rapid dissemination (Figure 1) [40,44–49]. The characteristics of the virus and the disease it causes are broadly representative of coronaviruses, and its pathological effects are highly similar to those of the previous two significant outbreaks—SARS and MERS.

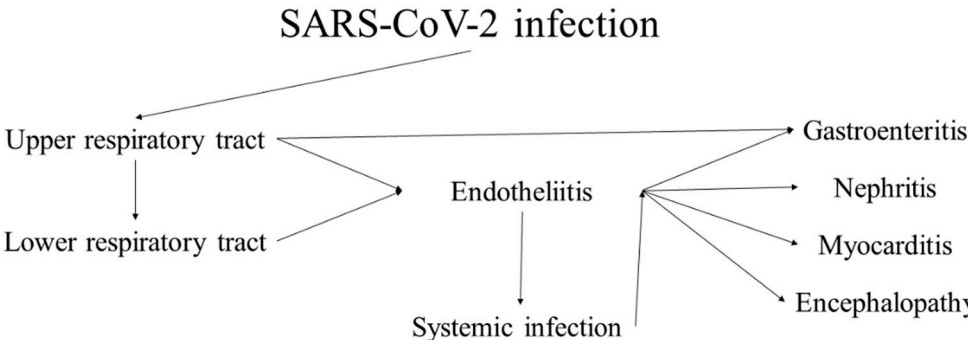

**Figure 1.** Progression and sequelae of SARS-CoV-2 infection (COVID-19).

## 2. Applications and Influences

Knowledge of the viral mechanism established in close relatives and the systemic effects of the infection are essential in caring for and treating ill patients, especially those in critical condition. Focus purely on the most severely affected system, i.e., the respiratory system, even with optimal care, while negating the effects of the virus on other systems may lead to rapid deterioration due to multisystem infection and the death of the patients, even in the absence of severe pathological effects to the respiratory system.

## 3. Coronaviruses and the Kidney

As already mentioned, SARS-CoV-2 is not the first entry of coronaviruses to cause an outbreak with severe health concerns in the last two decades. However, SARS-CoV-2 follows the same structural organization and similarities for tropism not only with other humans but also with animal coronaviruses that have yet to jump to human hosts. Severe acute respiratory syndrome coronavirus (SARS-CoV) and Middle Eastern respiratory

syndrome coronavirus (MERS-CoV), although much less studied, are close relatives of SARS-CoV-2, and looking back at the limited reports of those infections gave some of the first hints and the possible multisystem involvement caused by these viruses and the potential health complications [33,50,51]. Furthermore, these historical data also underline the systemic effects of coronaviruses, not only the respiratory ones, but also their renal tropism, as one of the earliest ways of cultivating them was using human embryonal kidney cell cultures.

### 3.1. SARS-CoV

The first identified coronavirus entry of concern, SARS-CoV, began its spread in 2002 as an infection in southern China [13]. The clinical course of the disease was severe, with predominantly respiratory symptoms and a mortality rate of 11% [34,52]. A total of just over 8000 cases have been identified, of which almost 800 are fatal, with the highest mortality rate noted in the age groups over 65 years old, where it is closer to 55%, and in all age groups, the prognosis is worse in men [52,53].

Thankfully, no new disease cases have been reported worldwide since 2004, with the last reported cases linked to intra-laboratory transmission in the virus study [54].

Explainable given the small number of infected patients and deaths worldwide, data on morphological changes caused by SARS-CoV are limited, and most of them focus on the lung parenchyma changes. In a report of eight autopsy cases, Franks et al. showed morphologic changes of diffuse alveolar damage clinically correlated with patients' main clinical features of acute respiratory distress syndrome, namely pulmonary hyaline membranes, hyperplasia of type 2 pneumocytes with multinucleation, macrophage infiltration with multinucleation, squamous metaplasia, and in patients surviving more than ten days after onset of the symptomatology and organization with interstitial fibroblastic proliferation, morphological changes virtually identical to those caused by SARS-CoV-2 [55].

Lang et al., for their part, reported three autopsy cases with gross hemorrhagic areas in the subpleural aspects of the lungs, as well as a histological finding represented by intestinal lymphoplasmacytic infiltration, capillary microthrombi, fibrin thrombi in larger vessels and diffuse alveolar damage with hyaline membranes, which the authors interpret as hemorrhagic inflammation, due to erythrocytic effusion in the alveolar spaces [56]. Other, albeit non-specific, changes have been described in the spleen with follicular atrophy and acute hemorrhage [56]. Again in a report of three autopsy cases, Ding et al. reported gross consolidation of the lung parenchyma, with acute focal hemorrhages and necrotic areas [57]. The histological finding is comparable to that described so far, except for intracytoplasmic inclusions in the alveolar epithelium, which are positive for Macchiavello staining [57]. A fundamental contribution of Ding's report is the description of the extrapulmonary manifestations of the infection, which confirm follicular atrophy and acute hemorrhages in the splenic parenchyma. On the side of changes in the organs not described in the previous two reports, the authors focus on the endothelial cells in the venous vessels of the heart, which are swollen, and the vessel walls are edematous with lymphocytic infiltration, a finding that we can interpret as endotheliitis–vasculitis, without a significant morphological finding in cardiomyocytes themselves [57]. In the liver, steatosis was described, and centroacinar necrosis was extensive in two cases, as well as perivascular lymphocytic infiltration around venous vessels [57]. In the brain parenchyma, in two of the cases, in addition to advanced cerebral edema with neuronal degeneration, the authors also described perivenous vasculitis–encephalitis with focal demyelination; in the muscles, myositis of the lower limbs was established with degenerative changes and lymphocytic inflammatory infiltration in and around the small arteries and veins [57]. In the bone marrow, changes with reduction in parenchyma were described, mainly due to the decrease in granulocyte–megakaryocytic lineages and background proliferation of polychromatophilic erythroblasts [57]. Areas of necrosis and a mononuclear inflammatory infiltrate—lymphocytes and macrophages, associated with venous blood vessels as well as

independently scattered in the interstitium of the kidney—with similar changes noted in the adrenal glands as well [57].

Although obtained from a small number of autopsy cases, the three independent reports show similarities in the type of lung and splenic damage, with Ding's report also showing the potential for extrapulmonary multiorgan involvement in the endotheliitis-venous vasculitis type and interstitial nephritis with necrosis [57]. Further research underlined the viral tropism to three main sites—the respiratory, gastrointestinal and urinary system—with the virus successfully replicating in the epithelial cells of the proximal tubules in the kidney, with the results being replicated ex vivo, leading to acute kidney injury with acute tubular necrosis in around 10% of cases [58–62]. However, those who developed acute kidney injury had an almost always lethal outcome [60,61].

Despite the limited literature data, the histopathology of SARS is virtually identical to that induced by SARS-CoV-2 [55–57,63–66].

### 3.2. MERS-CoV

The next known representative of coronaviruses causing severe clinical disease, MERS, began its spread in 2012 in Saudi Arabia [67,68]. The symptoms are similar to the previous representative with a severe clinical course. Unlike the interrupted epidemic chain of SARS, cases of MERS continue to be diagnosed to this day, and the main reason for this is the substantial spread in the natural reservoir of the infection—camels [69–71]. The main reported cases of infection transmission are primarily associated with the medical care of critically ill patients, indicating a low anthropogenic transmission potential of the virus [72–74].

The continuation of the epidemic chain also leads to new variations of the initially described virus, such as the initially described variant called Clades A (EMC/2012 and Jordan-N3/2012), while in recent years, a new variant called Clades B has emerged which, although genotypically different, leads to a similar course of the disease [75,76].

To date, nearly 3000 cases have been reported, of which nearly 900 have been fatal [77].

Again, due to the small number of cases, there are extremely few morphological reports of organ changes in cases with fatal MERS. In a report of a deceased 33-year-old male patient with T-cell lymphoma, Alsaad used an innovative technique for postmortem tissue examination by means of through-cut necropsy of lung, kidney, liver, heart, brain, and peripheral muscles taken 45 min after death [78]. Here, it is worth noting that the used methodology, although not perfect in terms of the completeness of the morphological examination, is innovative and simultaneously applicable and significantly safer in the necropsy examination of deceased patients from infectious diseases with unclear pathogenesis, organ damage, high contagiousness index and/or mortality—hazard groups three and four [79,80]. The morphological findings in the lungs show relative similarity to the lesions in SARS—diffuse alveolar damage, fibrin effusion in the alveolar spaces, interstitial infiltration by B lymphocytes and macrophages, parenchymal necrosis and subendothelial infiltration by T lymphocytes in medium-sized vessels [78]. In terms of liver changes, in addition to chronic portal hepatitis, centroacinar necrosis with polymorphonuclear infiltration was also found [78]. Degenerative-inflammatory changes with lymphocytic infiltration in the endo- and perimysium were found in the skeletal muscles [78]. No morphological changes were detected in the heart and brain [78]. In the renal parenchyma, the changes clinically correlate with acute renal failure (acute tubular injury)—dilatation of the proximal tubules with advanced cellular edema [78]. By means of electron microscopy, Alsaad also established the presence of virions in the cytoplasm and close to the nuclear membrane in pneumocytes, macrophages in the lung and muscle, as well as in the epithelial cells of the proximal segment of the renal tubules, showing a tropism of infection, not only to the lung but also to the kidney [78].

In another autopsy case report published by Ng, performed on a 45-year-old obese male, a different technique for decreasing the infectious risk of postmortem examination was applied [81]. To minimize the risk of transmission to the medical staff, the autopsy was

performed ten days postmortem, during which the deceased's body was stored at 4 degrees Celsius [81]. The report also describes the gross organ findings, including 5 L of pleural effusion, 150 mL of pericardial effusion and an unspecified amount of ascites fluid, without describing the characteristics of the effusion itself (serous, fibrinous, purulent, hemorrhagic, etc.) [81]. The lungs were grossly enlarged and heavy, diffusely consolidated [81]. The morphology of lung injury is comparable to that described in Alsaad's report, with an acute phase of diffuse alveolar damage—interalveolar edema, fibrin effusion with the formation of pulmonary hyaline membranes, thickened alveolar septa, hyperplasia of type 2 pneumocytes, multinuclear syncytial cells and mixed interstitial and peribronchial infiltrate [81]. Viral particles were detected via immunohistochemistry, mainly in epithelial cells of bronchial glands and pneumocytes and macrophages [81]. Severe hypertensive changes with focal interstitial inflammation were described in the kidney [81]. No specific changes were described in the lymph nodes (reactive changes), spleen (reactive changes), bone marrow (preserved hematopoiesis with increased granulopoiesis), heart (hypertensive heart) and central nervous system [81]. In the liver, steatosis, focal calcifications and portal hepatitis were described [81]. Except for the lung parenchyma, no viral particles were detected via immunohistochemistry [81].

From a clinical perspective, more than 50% of critically ill cases develop acute kidney injury, requiring renal replacement therapy [82]. Other than direct tubular cell infection and necrosis, it has also been suggested that the effects of MERS-CoV on the kidney are also due to the induction of apoptosis in these cells [83–85].

There are also data on the morphology of chronic renal consequences of MERS in a patient who underwent a kidney biopsy eight weeks after infection due to newly appearing symptoms [85]. Morphologically, the changes were represented by acute tubular necrosis and sclerosis with protein casts and tubulointerstitial nephritis without glomerulosclerosis present [85]. However, no viral particles were detected in the biopsy [85].

Once again, as with SARS, MERS-induced histopathological changes are virtually identical to those induced by SARS-CoV-2 [55,56,63,78,81].

### 3.3. SARS-CoV-2

The literature is increasingly enriched with data on morphological changes in the kidney during and after SARS-CoV-2 infection, given the high frequency of clinically reported renal complications, including acute kidney injury and chronic kidney failure developing after infection [48,86,87].

At the beginning of the pandemic, initial reports on kidney morphology, predominantly case reports and small cohort studies, reported changes akin to tubulointerstitial nephritis—lymphocytic infiltration and tubular degeneration [66,87].

Evidence suggests that acute kidney injury is clinically and morphologically observed in about 50% of severe and fatal cases [88]. Most often, morphological changes are associated with acute tubular injury with epithelial swelling and necrosis, interstitial lymphocyte and macrophage infiltration and an admixture of cast nephropathy phenomena, with electron microscopy, also suggesting that there is active viral replication within the tubular epithelial cells leading to their degeneration and necrosis and inducing nephritis [87–89]. Interstitial changes reported in around 10% of fatal cases are also thrombosis, endotheliitis, cortical necrosis and infarctions, as well as myoglobin casts due to rhabdomyolysis [88]. From the glomerular aspects of the kidney injury, fatal cases often show predisposing injury such as hypertensive and diabetic nephropathy; however, it is interesting that these cases also show a significant percentage, varying around 10% as based on most studies, with thrombotic microangiopathy—glomerular capillary loop fibrin thrombi [88,90]. Interestingly, thrombotic microangiopathy in these patients develops in the context of active anticoagulant treatment, and it is suggested that without the administration of anticoagulants, a significantly higher percentage of patients with COVID-19 would develop this morphological phenomenon of glomerular injury [86,87,90]. Electron microscopy of renal tissue samples also suggests direct podocyte injury and direct viral replication in them,

leading to podocyte feet shortening, desquamation and subsequent hyperplasia—a collapsing glomerulopathy form of focal segmental glomerulosclerosis, which to a large extent, in addition to the vascular changes, contributes significantly to the decreased renal function and kidney injury onset [89,91–96]. Other than the collapsing glomerulopathy form of focal segmental glomerulosclerosis, cases of pauci-immune crescentic glomerulonephritis have also been described [97].Thankfully, most patients who survive show at least moderate improvement in kidney function after COVID-19 [90]. In our practice, we have encountered patients that have previously had no kidney-related health concerns but several weeks after a mild clinical form of COVID-19 saw a rapid decrease in kidney function and required dialysis, with several other reports also depicting similar disease courses [90].

Other reported types of non-specific injury to the parenchyma, even in the context of no previous clinical signs or symptoms, have broadly been accepted as a previously unrecognized co-morbidity and include varying types of glomerulonephritis and amyloidosis [87,88,90,91].

In addition to the native kidney, data are also available for acute injury in transplanted kidneys with antibody-mediated rejection developing several weeks after COVID-19 [90,98]. In a small sample, despite most of the patients not having a severe form of COVID-19, allograft biopsy showed rejection phenomena—acute and chronic, as well as acute tubular injury and collapsing, perihilar, tip-lesion and focal segmental glomerulosclerosis, with some patients requiring dialysis due to the allograft injury [98].

### 3.4. COVAN

The accumulated data on COVID-19 patients, combined with the limited data available on patients with SARS and MERS, underline not only the presence of an epitheliotropism in this viral family but also the ability of severe clinical variants to manifest not as an organ-specific and limited disease, but as an actual multisystem disease. As renal involvement is relatively common in the course of COVID-19 and has a significant morphological substrate COVID-19-associated nephropathy, COVAN has been a suggested nomenclature addition to signify not only the possible acute but also chronic sequelae in these patients, especially in preexisting renal diseases (Figure 2) [48,99–102]. However, renal involvement is not universal and varies in severity; a genetic predisposition has also been suggested in people with African ancestry and APOL1 genotype [99].

## COVAN

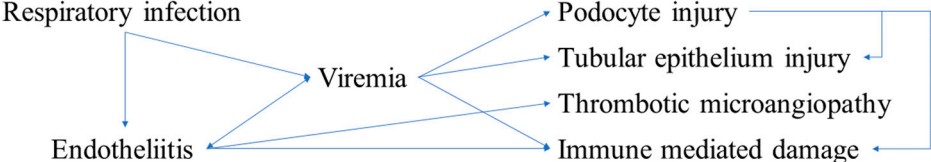

**Figure 2.** Development and components of COVAN.

In retrospect, COVAN, despite being discussed only in narrow fields and poorly presented to the general public, is a logical component of the disease. The presence of direct viral replication in renal epithelial cells explains the presence and a concentration correlated with the epidemiological process of SARS-CoV-2 RNA in municipal wastewater, while solid fracture wastewater concentration, while also probably correlated with COVAN, is more indicative of gastrointestinal involvement as well [103,104].

## 4. Conclusions and Prospects

- Coronaviruses are epitheliotropic viruses, with three variants of concern emerging over the last two decades;
- While respiratory symptoms dominate these diseases' clinical course, many other systems and organs are also a direct viral target;

- One of these is the kidney. The renal involvement, designated as COVAN, is due to:
  - ○ direct viral replication and damage to the podocytes;
  - ○ Direct viral replication and damage to tubule epithelial cells, resulting in:
    - ■ glomerulopathy and tubule-interstitial nephritis with acute kidney injury, latent kidney injury and chronic kidney injury, requiring dialysis in such patients;
- One of the main non-direct components of COVAN is the development of thrombotic microangiopathy, even in the context of active anticoagulant treatments, indicating a two-hit mechanism;
- Awareness of these complications, active monitoring and preventive as well as systemic treatment will inevitably decrease mortality and improve life quality in the context of post-COVID-19 syndrome.

**Author Contributions:** H.P. and G.S.S. conceptualized the study; H.P. and G.S.S. performed initial literature review; H.P. performed detailed review on historical data; H.P., A.B.T., M.I., D.S. and L.P. performed review on morphology and mechanisms; G.S.S., M.I., D.S. and L.P. conceptualized and produced the figures; H.P., G.S.S. and L.P. wrote the initial manuscript; A.B.T., D.S. and M.I. performed critical revisions; A.B.T. revised and approved the final version of the manuscript for submitting; H.P., G.S.S. and A.B.T. performed revisions after review. All authors have read and approve the final version of the published manuscript.

**Funding:** This research received no external funding.

**Institutional Review Board Statement:** Not applicable.

**Data Availability Statement:** Not applicable.

**Conflicts of Interest:** The authors declare no conflict of interest.

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
