# Peer review of "SARS, MERS and COVID-19-Associated Renal Pathology"

_encyclopedia, doi:10.3390/encyclopedia2040117_

Round 1

Reviewer 1 Report

- The definition is incorrect. For example, the sentence - "Coronaviruses are a large group of viruses, the most notable of which are SARS-CoV, MERS-CoV and SARS-CoV-2." it is non-informative and there is no place for such sentences in the definition.

- The definition applies to SARS, MERS and SARS-CoV-2 - associated renal pathology, so it should contain definitions of disease entities and their infectious agents with the most essential condensed information, and it cannot be just any loose story. Suggested ingredients: coronaviruses - what group, who and when was it detected, pathology, frequency of occurrence; SARS - a definition of who and when detected an infectious agent, general pathology and in detail the essence and significance of renal pathology / COVAN with the incidence, complications and prognosis, significant differences with other pathogenic coronaviruses. The same for MERS and COVID-19.

- The rest of the text should also be more structured as encyclopedic, indicating what data we do not have yet and which would be important in the area of ​​renal pathology.

Author Response

  • The definition is incorrect. For example, the sentence - "Coronaviruses are a large group of viruses, the most notable of which are SARS-CoV, MERS-CoV and SARS-CoV-2." it is non-informative and there is no place for such sentences in the definition.

Dear reviewer, we have followed the requirements for entry structure as provided by the publishing house, which state, "Please provide a brief overview of the entry title. Defining explanations should focus on the most essential points of information about the topic being described". By starting the definition with the specified statement, we indent to introduce the reader to the topic softly. As encyclopedias are used as a general source of information, this is intended more for the general reader and not for medical professionals and practitioners. We have introduced modifications to this section wherever possible without contradicting the publishers' guide.

  • The definition applies to SARS, MERS and SARS-CoV-2 - associated renal pathology, so it should contain definitions of disease entities and their infectious agents with the most essential condensed information, and it cannot be just any loose story. Suggested ingredients: coronaviruses - what group, who and when was it detected, pathology, frequency of occurrence; SARS - a definition of who and when detected an infectious agent, general pathology and in detail the essence and significance of renal pathology / COVAN with the incidence, complications and prognosis, significant differences with other pathogenic coronaviruses. The same for MERS and COVID-19.

As stated in the previous section of our reply, we have followed the designated instructions provided by the publisher for this section. This section should serve as the manuscript's abstract and include condensed information; hence, the suggested data would make it too long and deter the general reader. The suggested data furthermore is presented in the main body of the manuscript in its specified subsections and present the data available in the medical literature. As stated in the previous section of our response, we have introduced changes in the definition section to modify it as best as possible without contradicting the publishers' notes.

  • The rest of the text should also be more structured as encyclopedic, indicating what data we do not have yet and which would be important in the area of ​​renal pathology

For the remainder of the manuscript, we have also used the template and the instructions provided by the publisher, which define an entry manuscript as "an educational material that offers a wider view of an entire research domain". It is our understanding that hence entry manuscripts for the encyclopedia should provide the information available on the topic within the scientific literature, rather than using the stereotypical "more research in deeded in this aspect to fully understand it" statement, which has widely become used to justify the study limitations nowadays in original works.

With this in mind, we have modified the manuscript to your suggestion, where it is not contradictory to the instructions of the publisher.

Reviewer 2 Report

In their contribution to this entry, the authors examined renal pathology associated with SARS, MERS and COVID-19. Considering the profound effort of these viruses, especially COVID-19 have on every aspect of daily life, with the after effects of COVID-19 just emerging, this entry is of current interest.

Overall this entry is very well-written and informative. The structure of this entry is sound and logical, the sections flowing seamlessly from one to the next.

The only comment is for the paragraph on SARS-CoV (section 3.1), there is very little information on the renal pathology despite very detailed description on lungs etc. Considering the title of the entry indicated that the focus is on renal pathology, more focus on the renal aspect on SARS-CoV would be appropriate.

Author Response

  • In their contribution to this entry, the authors examined renal pathology associated with SARS, MERS and COVID-19. Considering the profound effort of these viruses, especially COVID-19 have on every aspect of daily life, with the after effects of COVID-19 just emerging, this entry is of current interest.

Thank you for this statement!

  • Overall this entry is very well-written and informative. The structure of this entry is sound and logical, the sections flowing seamlessly from one to the next.

Thank you for this statement!

  • The only comment is for the paragraph on SARS-CoV (section 3.1), there is very little information on the renal pathology despite very detailed description on lungs etc. Considering the title of the entry indicated that the focus is on renal pathology, more focus on the renal aspect on SARS-CoV would be appropriate.

Thank you for this suggestion! Sadly, there are only a few autopsy cases, so information on this aspect is limited. However, we have expanded the statements to include as much of the information depicted by the original authors as possible.

Round 2

Reviewer 1 Report

Acknowledges the authors' responses to the suggestions. The changes made are right.
The condensation of the information given could be greater, but the present form is sufficient for publication.

Author Response

  • Acknowledges the authors' responses to the suggestions. The changes made are right.

Thank you for this statement about the construction of our manuscript.

  • The condensation of the information given could be greater, but the present form is sufficient for publication.

Thank you for this statement! We have further condensed the information in the conclusion section of the entry.

Reviewer 2 Report

The reviewer would like to thank the authors for the additional paragraphs, which makes the entry more complete. The only other comment is the name of genes should be in italics:.... APOL1 genotype (Line 339).

Author Response

  • The reviewer would like to thank the authors for the additional paragraphs, which makes the entry more complete. The only other comment is the name of genes should be in italics:.... APOL1 genotype (Line 339).

Thank you for this correction suggestion; the change has been implemented.

Round 3

Reviewer 2 Report

A detailed, comprehensive and well-written entry on renal pathology associated with SARS, MERS and COVID-19. Congratulations!